# The Possibility of Separation of Heavy Minerals as Byproduct of the Danube River Gravel Sand Extraction

**Michal Maťašovský [1], Martin Sisol [1], Michal Marcin [1,*] and Peter Uhlík [2]**

1 Institute of Earth Resources, Faculty of Mining, Ecology, Process Control and Geotechnologies, Technical University of Košice, Letná 9, 040 01 Košice, Slovakia; mmatasovsky@gmail.com (M.M.); martin.sisol@tuke.sk (M.S.)

2 Department of Mineralogy, Petrology and Economic Geology, Faculty of Natural Sciences, Comenius University Bratislava, Mlynská dolina, Ilkovičova 6, 842 15 Bratislava, Slovakia; peter.uhlik@uniba.sk

* Correspondence: michal.marcin@tuke.sk

**Abstract:** The history of exploitation of gold from the Danube River's sandy gravels is centuries long. The extraction of valuable heavy minerals (VHM) concentrate was never intended. Our aim is to find out an effective separation process to produce monomineral concentrates of the following minerals: garnet, ilmenite, zircon, monazite, magnetite, rutile, gold. The essential condition is to use no chemistry (no flotation, leaching, activating). The experimental concentrates were prepared by sluicing on active river channel. Next, the separation results were achieved using gravity and electromagnetic methods with different magnetic intensities. The prepared rutile contained from 63.3% $TiO_2$ to 87% $TiO_2$. The ilmenite concentrate contained 20.5% $TiO_2$ and 39.2% ilmenite. The garnet concentrate contained 94% garnet. The monazite concentrate contained 86.6% monazite, and the sum of REE oxides was 50.1%. The zircon concentrate containing 63.7% $ZrO_2$ means that the prepared concentrate contained 96.1% zircon.

**Keywords:** critical raw materials; gravelly sand; gravity and electromagnetic separation; heavy mineral concentrates; monazite; gold; zircon; rutile; ilmenite and garnet





## 1. Introduction

Even though metal recycling is becoming a worldwide trend, it still falls short of meeting the demand due to low recycling rates. Furthermore, there are no efficient alternatives to some metals, such as titanium, rare earth elements (REE), zirconium and hafnium. Therefore, new exploration resources must be discovered and exploited in order to obtain these minerals [1,2]. Furthermore, certain modern technologies (electronic devices, electric vehicles) necessitate the use of specific metals, without which these technologies would not have progressed as far as they have. As the European Union is dependent on the import of raw materials, some of them have been recognized by the European Commission as critical (e.g., REE, Ti) [3,4]. As a result, heavy minerals, such as monazite, rutile, ilmenite, are regarded as important [2]. Heavy minerals (HM) are minerals (rock forming, accessory, i.e., ore minerals) with high specific gravity (SG > 2.9 g/cm³). As they are weather resistant, such as rutile, magnetite, ilmenite, zircon, monazite, garnets, cassiterite, tourmalines, pyroxenes, and staurolite, these minerals are concentrated in siliciclastic sediments. Due to their utility in industrial goods, such as pigments (ilmenite, rutile and leucoxene), ceramics (zircon), abrasives (garnet) or in the recovery of high-value components, such as rare earth oxides, these heavy minerals have been recognized as valuable heavy minerals (VHM). Valuable heavy minerals (VHM) could be recovered for commercial use as marketable products if the individual product concentrations matched market standards [5,6].

Valuable heavy minerals (VHM) are abundant in the finest particles of sands and form placer deposits, which are formed in alluvial, marine and aeolian environments [7–13]. Additionally, while sands and placer deposits may have a low concentration of heavy

minerals when compared to other mineral sources, the vast amount of these makes them potentially appealing for processing [9]. Valuable heavy minerals have been collected and isolated from placer deposits, mainly from marine beach sands [2,5–12]. Frihy et al. [11] proved that dredged sediments with a concentration of heavy minerals of more than 2 wt % percent are economically profitable.

The majority of beneficiation of sands or aggregates starts with a gravity separation to separate heavy minerals from gangue, which is mostly quartz [13–15]. Comminution is usually not required for these minerals, which reduces processing costs and saves energy [2]. Depending on the properties of each mineral to be separated, heavy minerals can be separated by magnetic separation, electrostatic separation or flotation [10,12–16]. Advanced separator devices, such as fluidized hydrocyclone separator or Falcon concentrator, are proved to be useful for selective separating of heavy minerals [17,18]. As a result, it is critical to choose a mineral processing method that will yield the greatest VHM while still being quick and efficient.

The aim of the presented work is to design a technological process of treatment of heavy mineral concentrates, which can be obtained as a byproduct during the extraction and processing of sandy gravels, i.e., riverbed maintenance and, at the same time, estimating the raw material potential of the areas of interest. In this study, the Danube River alluvial sandy gravels from Slovakia were used as a feed sample. The beneficiation process, especially magnetic separation, and mineralogy were applied to separate valuable mineral selectively and examine the feasibility of a resource development project.

## 2. Materials and Methods

The sands used in this work were obtained from the beaches of the Danube main river channel at approximately 1778 river kilometer, in the southwestern part of Slovakia, near the town of Komárno. For the preparation of the heavy mineral concentrate, a riffle sluice with a total length of 2.8 m was used (Figure 1). The width of the sluice was 41 cm, with the lower two-thirds gradually tapering from a width of 41 to 30 cm. The 1 m × 0.4 m hopper was fitted with a 4 mm × 6 mm mesh slotted screen. The sluice was lined with carpeted partitions arranged in groups of three, with approximately 5 mm between each partition and 1 cm between groups of partitions (Figure 1). The longitudinal slope of the riffle sluice of 5 to 9 degrees was adapted to the amount of incoming water (3.7 to 5.6 $m^3 \cdot h^{-1}$), which depended on the type of slurry pump used. One ripping cycle lasted approximately two hours. During the sluicing cycle, the bottom carpet of the sluice was washed at approximately 5–10 min intervals whenever the baffles were filled with concentrate. At the end of the cycle, all carpets were washed into a prepared container.

The heavy mineral concentrates thus obtained were subsequently dried and sorted to grain size classes above and below 0.315 mm. The grain size class above 0.315 mm was considered waste; grain size below 0.315 mm contained almost 99% of the heavy minerals. The fraction below 0.315 mm formed the input heavy mineral concentrate—the feed for the following experiments.

### 2.1. Gravity Concentration

Gravitational separation experiments of the prepared heavy mineral concentrates were carried out on a concentration table.

The homogenized batch (sorted heavy mineral concentrate below 0.315 mm for the Danube alluvial) was distributed on a Jiangxi Jinshibao Mining Machinery Manufacturing Co., Ltd. (Ganzhou, China) shaking table, with a work surface of 210 cm × 106 cm on the drive side and 86 cm on the yield side, respectively. The total surface area of the work intended for medium- to coarse-grained materials was 2.02 $m^2$, the length of the riffles was 22 mm, and the number of strokes was 210 per minute. The inclination of the work surface was 2–3°. The amount of solid particles in the feed was 25–30%.

The gold was extracted from the obtained concentrates using a pan and The Thumper—a portable wave table.

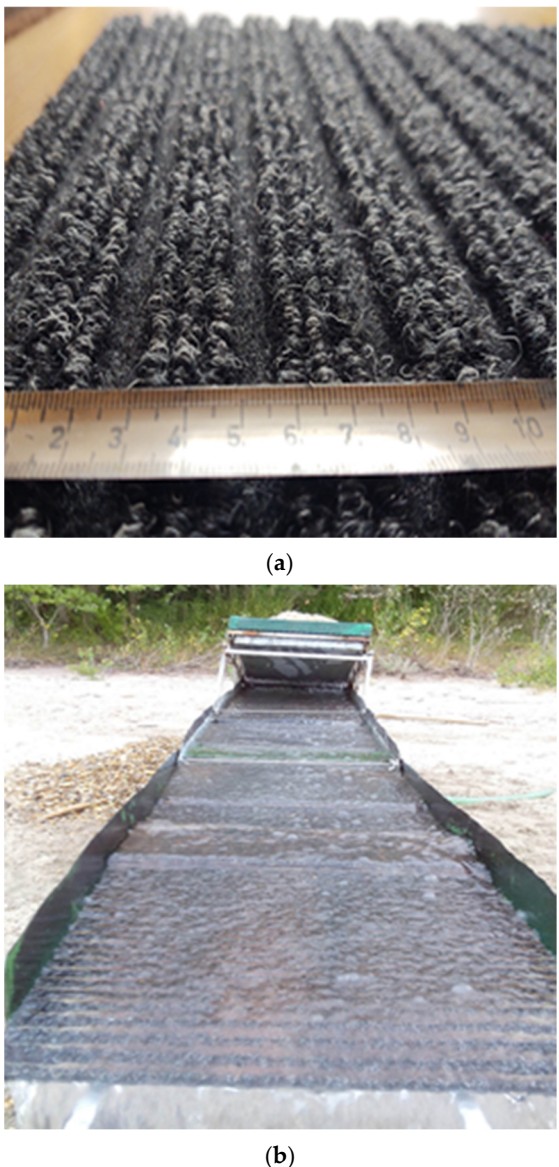

(**a**)

(**b**)

**Figure 1.** Detailed picture of the wash carpet (**a**). Overlook at the riffle sluice (**b**).

*2.2. Electromagnetic Separation*

The intermediate product of the gravity treatment of heavy mineral concentrate from the gravel sands of the Danube Alluvium was electromagnetically separated on a semi-operational electromagnetic separator Mechanobr with a rotating pole and an electrome-chanical surge feeder. The batch was fed from the hopper by means of a drum feeder and evenly distributed on a moving rubber belt. The magnetic poles were located above a moving rubber belt with a batch on a rotating disk extending beyond the edges of the belt. The concentrate and intermediate product were thus discharged into containers on both sides of the belt. The non-magnetic fraction was carried to the hopper behind the return drum of the rubber belt. The applied magnetic intensity was increased from 0.25 T to 1.4 T. After recovering magnetic products at 0.25 T, the remaining product was fed into a magnetic separator that was adjusted to a higher magnetic intensity for further separation. The magnetic separation tests were carried out sequentially in this way until the magnetic intensity reached 1.4 T.

### 2.3. Mineralogy of VHM

The fine aggregate waste and concentrates were analyzed by X-ray powder diffraction (XRD) to determine their qualitative and quantitative composition. Most samples were analyzed by the diffractometer (XRD; Bruker D2 Phaser, Mannheim, Germany). The XRD patterns of the samples were obtained under the following conditions: CuKα radiation, monochromatic Ni-filter, accelerating voltage of the X-ray generator 30 kV, current intensity 10 mA, area of scan angles: 5–70° 2θ, time/step 0.3 s/0.01°. Processing and interpretation of the XRD record were realized with the Bruker software DIFFRAC EVA V3.1. (Bruker, Billerica, MA, USA) using an internal database of mineral standards Powder Diffraction Files PDF-2/2013 (by IUCr = Database COD) and control reference chart AMS data.

Quantitative mineralogical composition of the selected concentrations was identified by XRD with internal standard [19]. The powder sample was below 0.02 mm; a 1 g weight was mixed with 0.2 g of $Al_2O_3$ as an internal standard. This mixture was ground for 5 min in a McCrone mill for better homogenization of the sample. The randomly oriented, side-loading specimens were analyzed on a Philips PW 1710 diffractometer (Cu Kα radiation, voltage 35 kV, current 20 μA, step 0.02° 2Θ, exposure time 2 s/step using a graphite monochromator). The interval 4–65° 2Θ was measured. The XRD patterns were evaluated using the RockJock software [20]. RockJock determines the quantitative content of minerals in powdered samples by comparing the integrated reflection intensities of individual minerals with the intensities for pure standard minerals and an internal standard.

### 2.4. Chemical Analysis

Samples were pulverized to a grain size of 85% below 0.075 mm and subsequently analyzed according to accredited analytical procedures for individual mineral concentrates. The concentrates were analyzed by a combination of inductively coupled plasma mass spectrometry (ICP-MS), inductively coupled plasma atomic emission spectroscopy (ICP-AES) and X-ray fluorescence spectrometry (XRF). For magnetite and monazite concentrates, the thermogravimetric analyzer (TGA) was applied. Following the requirement to determine the U and Th content of the zircon concentrate, the U and Th content was determined by ICP-MS. Determination of the gold content was carried out by cupellation and atomic absorption spectroscopy (AAS).

## 3. Results

In the first phase, a pilot experiment was carried out, the results of which were published in Ref [21]. A heavy mineral concentrate with a total weight of 73.6 kg, obtained from 300 kg of the deposit (alluvial sandy gravels), was sorted into grain size fractions above 0.5 mm, 0.315–0.5 mm and below 0.315 mm. The fraction above 0.5 mm, weighing 3 kg, represented mainly by quartz and quartzite (85%), limestones and dolomite (10%), metamorphic rocks and sandstones (5%), constituted waste. The fraction 0.315–0.5 mm weighing 12.6 kg and the fraction below 0.315 mm weighing 58 kg were processed on a shaking table (Jiangxi) and on a magnetic separator (Mechanobr) with a gradual increase in magnetic field strength by increasing the value of the saturation current. The representation of the individual minerals in the heavy mineral concentrate in the grain size class 0–0.315 mm from the XRD is shown in Figure 2, and the mineralogical composition of the feed is shown in Table 1.

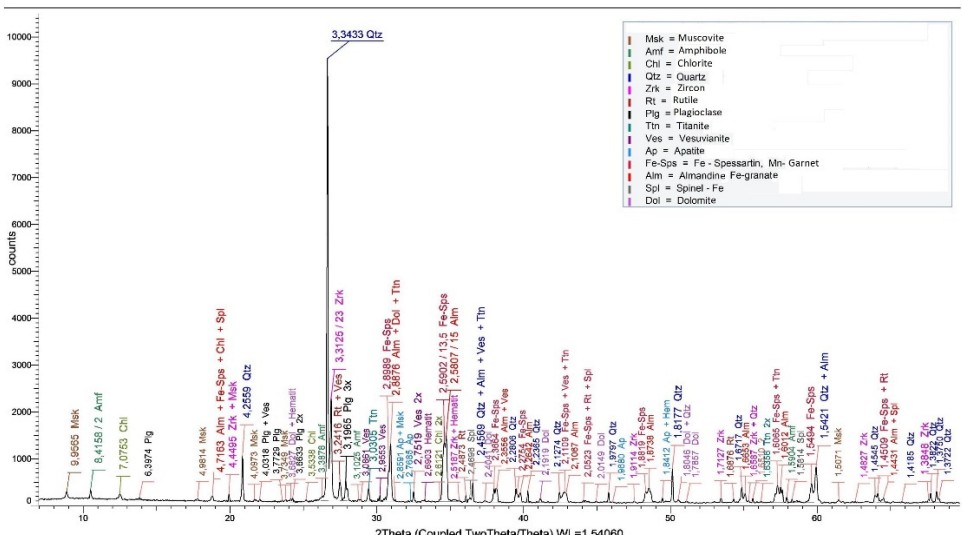

**Figure 2.** XRD diffractogram of the feed sample.

**Table 1.** Mineralogical composition of the feed sample.

| Mineral | wt % |
|---------|------|
| Quartz | 35.3 |
| Garnet | 31.4 |
| Zircon | 17.2 |
| Rutile | 5 |
| Dolomite | 3.9 |
| Titanite | 1.5 |
| Amphibole | 1.5 |
| Plagioclase | 1.4 |
| Vesuvianite | 1 |
| Hematite | 0.9 |
| Fe-Spinel | 0.8 |
| Chlorite | 0.5 |
| Apatite | 0.3 |

From the feed of the fraction below 0.315 mm, 41,200 g of heavy minerals was recovered, representing a weight yield of 71.03% of the grain size class, 55.98% from the heavy mineral concentrate and 13.73% of the feed (sandy gravels). The total heavy mineral content of the feed was thus 14.08%. Since 97.53% of the heavy minerals in the gravels was found to be in the grain size class below 0.315 mm, further experiments were carried out on the heavy mineral concentrates, which were sorted to the required grain size.

The feed for the main experiment was 1300 kg of the concentrate of heavy minerals sieved below 0.315 mm. The homogenized heavy mineral concentrate was processed on a shaking table. A simplified scheme of the whole experiment can be seen on the process flow diagram on Figure 3.

The products of the first and second steps of the gravity shaking table tests were combined, homogenized and dried. Subsequently, they were processed in several steps on an electromagnetic separator (Mechanobr) with a gradual increase in magnetic field strength. Increasing the value of the saturation electric current was carried out in the following sequence: excitation current (intensity of magnetic field in parentheses), 0.25A (0.275 T), 0.5A (0.320 T), 1A (0.580 T), 2A (1.06 T), 3A (1.19 T), 4A (1.27 T), 5A (1.31 T), 6A (1.35 T), 7A (1.39 T). The products of the electromagnetic separation were concentrates of rutile (MS 1A, MS 5A, MS 7A). XRD diffractograms are shown in Figure 4, and mineralogical analysis is shown in Tables 2–4. The increase in the use intensity of the magnetic field increased the rutile concentration from 80 to 92.6 wt %.

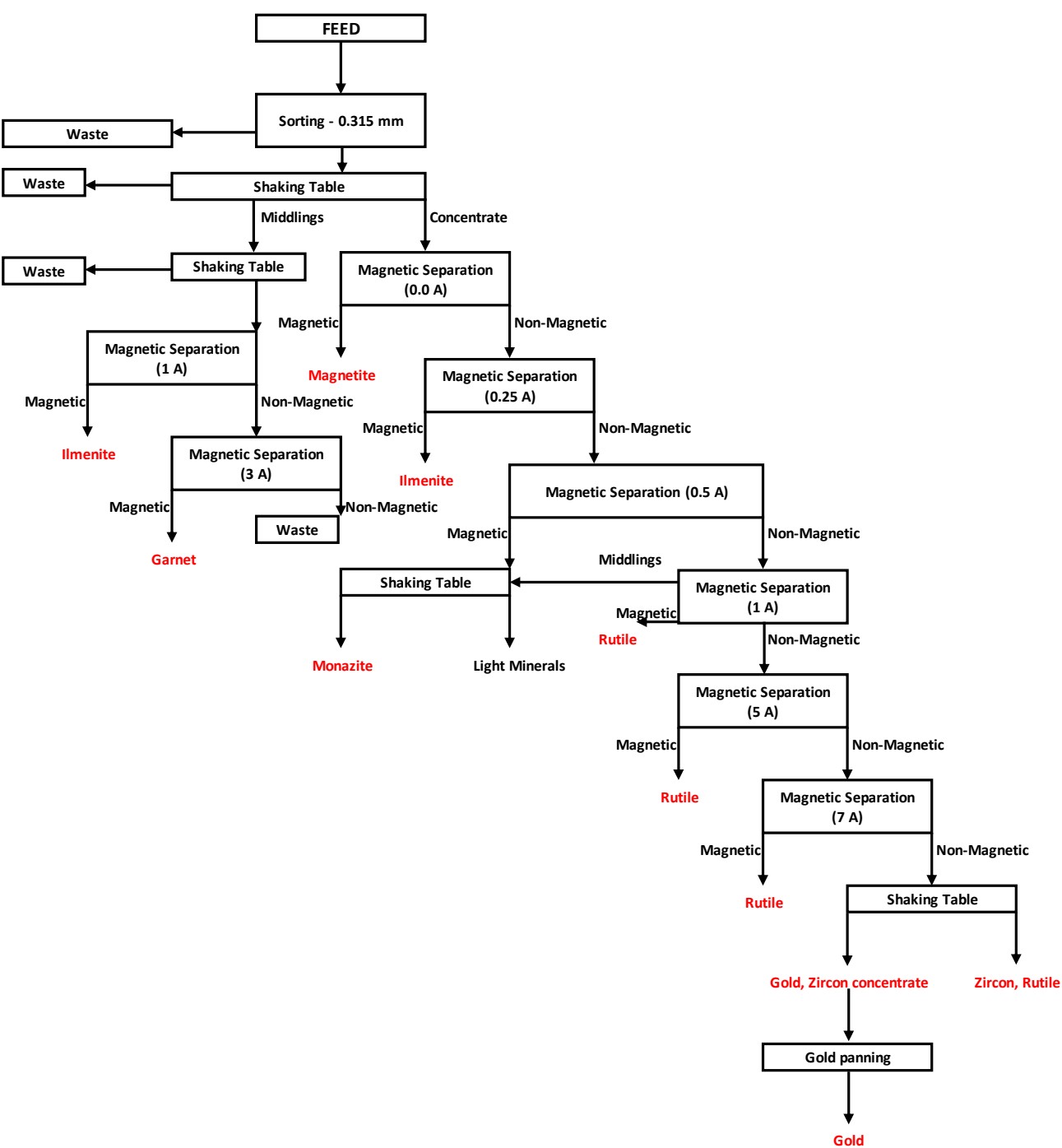

**Figure 3.** Scheme of processing VHM concentrates from the feed sample.

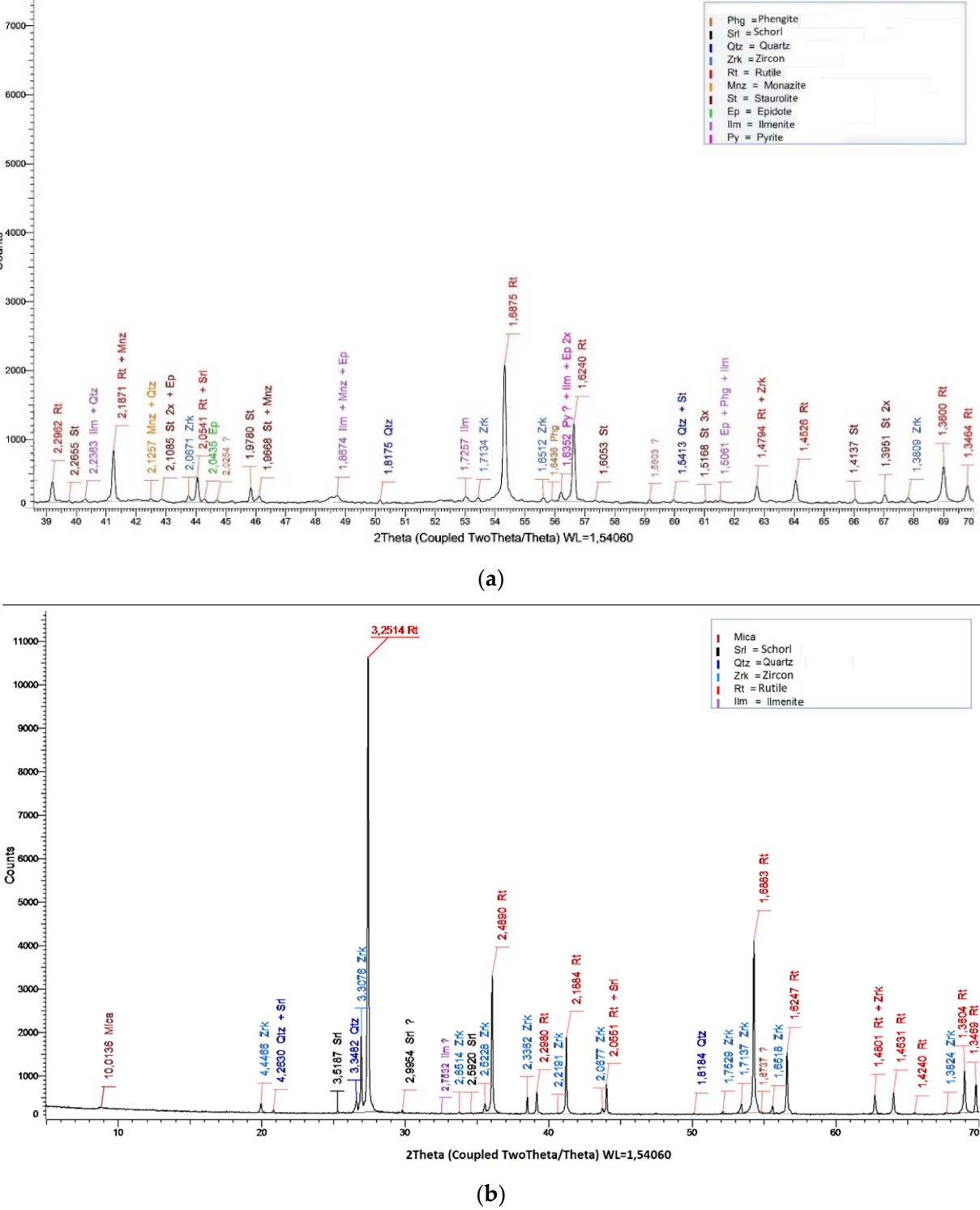

(**a**)

(**b**)

**Figure 4.** *Cont.*

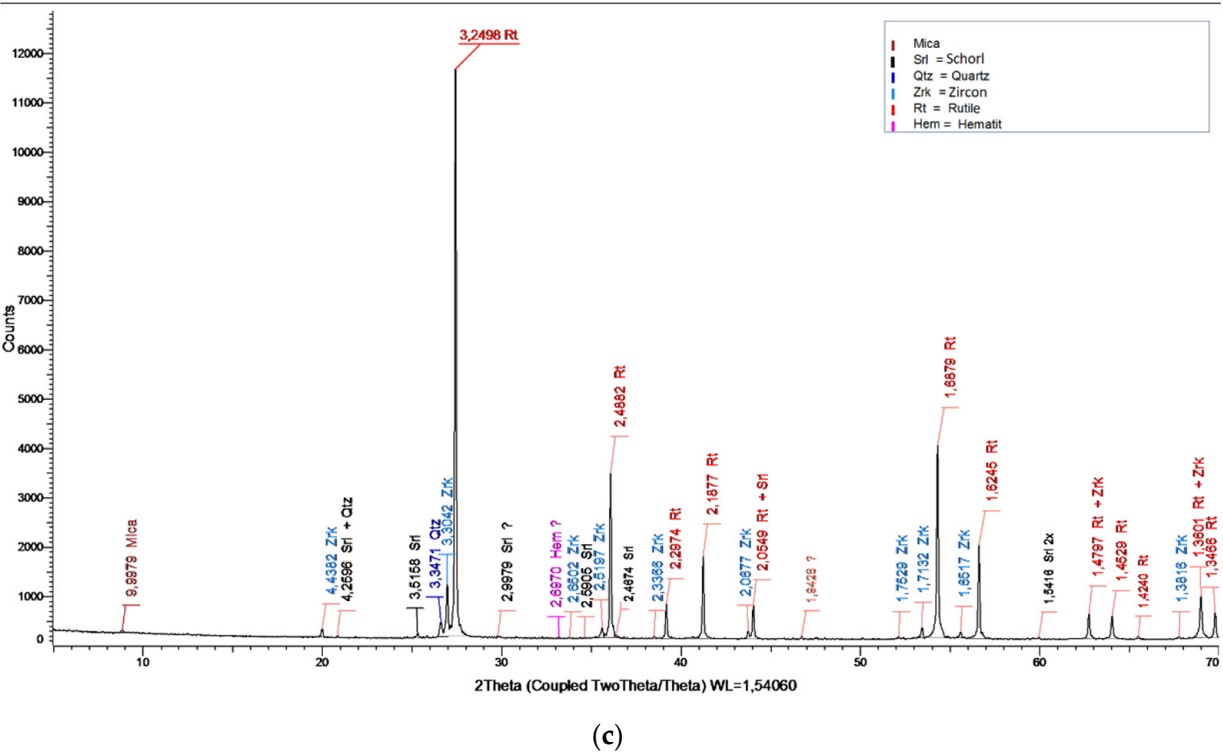

(**c**)

**Figure 4.** XRD diffractograms of rutile concentrate MS 1A (**a**), MS 5A (**b**), MS 7A (**c**).

**Table 2.** Mineralogical composition of rutile concentrate from MS 1A.

| Mineral | wt % |
|---|---|
| Rutile | 80 |
| Monazite | 6.8 |
| Ilmenite | 5.5 |
| Zircon | 2.6 |
| Hematite | 1.2 |
| Quartz | 0.8 |
| Epidote | 0.7 |
| Schorl | 0.5 |
| Phengite | 0.5 |
| Other | 1.4 |

**Table 3.** Mineralogical composition of rutile concentrate from MS 5A.

| Mineral | wt % |
|---|---|
| Rutile | 88.7 |
| Zircon | 9.7 |
| Quartz | 0.6 |
| Other | 0.6 |

**Table 4.** Mineralogical composition of rutile concentrate from MS 7A.

| Mineral | wt % |
|---|---|
| Rutile | 92.6 |
| Zircon | 5.6 |
| Quartz | 0.7 |
| Other | 1.0 |

Rutile concentrates prepared by electromagnetic separation at different levels of magnetic field strength contained from 63.3% $TiO_2$ (MS 1A) to 87% $TiO_2$ (MS 7A). Rutile concentrates can be a source of $TiO_2$ and titanium, respectively, with MS 7A concentrate achieving the quality parameters of commercial rutile concentrates [22].

The middlings from the first shaking table tests were processed on an electromagnetic separator (with a rotating disk) in two steps, increasing the value of the saturation current from 1A to 3A with an increase in magnetic field strength. The product was the ilmenite concentrate (1A) and garnet concentrate (3A). Mineralogical analysis of the product is shown in Tables 5 and 6.

**Table 5.** Mineralogical composition of ilmenite concentrate from MS 1A.

| Mineral | wt % |
| --- | --- |
| Hematite | 48.4 |
| Ilmenite | 39.2 |
| Garnet | 3.1 |
| Rutile | 2.3 |
| Quartz | 0.9 |
| Other | 6.1 |

**Table 6.** Mineralogical composition of garnet concentrate from MS 3A.

| Mineral | wt % |
| --- | --- |
| Garnet | 94.0 |
| Tourmaline | 2.1 |
| Quartz | 0.7 |
| Other | 3.2 |

The ilmenite concentrate contained 20.5% $TiO_2$ and 74.7% $Fe_2O_3$. According to mineralogical analysis, it contained 48.4% hematite and 39.2% ilmenite. XRD and chemical analyses show that a significant part of ilmenite in the concentrate corresponded to hemoilmenite and titanohematite, respectively, according to the classification [23]. After the cleansing operation, the concentrate could represent a feed to produce the synthetic rutile.

The garnet concentrate contained 94% garnet, according to mineralogical analysis. The general requirement for garnet content in the concentrate is more than 97% [22]. With a quartz content of 0.7% as a contaminating unwanted component, it meets the requirement of the standard (ISO 11126-10:2000) for garnet concentrates intended for waterjet cutting. According to this standard, the quartz content must not exceed 1% [22].

Monazite concentrate was obtained by gravity concentration on the shaking table from the magnetic products of 0.5A and 1A magnetic separation. XRD diffractogram of the concentrate is shown in Figure 5 and mineral composition in Table 7.

The monazite concentrate contained 86.6% monazite, according to the X-ray diffraction analysis. The sum of REE oxides, according to the chemical analysis, was 50.1%. The REE content of commercially available concentrates ranges from 55% to 58% [23]. The U and Th contents were 0.265% and 4.24%, respectively. The monazite concentrate potentially represents a source of REE, a critical mineral resource in the EU.

The non-magnetic fraction from the electromagnetic separation was reprocessed on a shaking table. The product was zircon middlings—gold concentrate and rutile. Gold was extracted from the zircon product using a gold panning process. Mineralogical analysis of zircon concentrate is shown in Table 8.

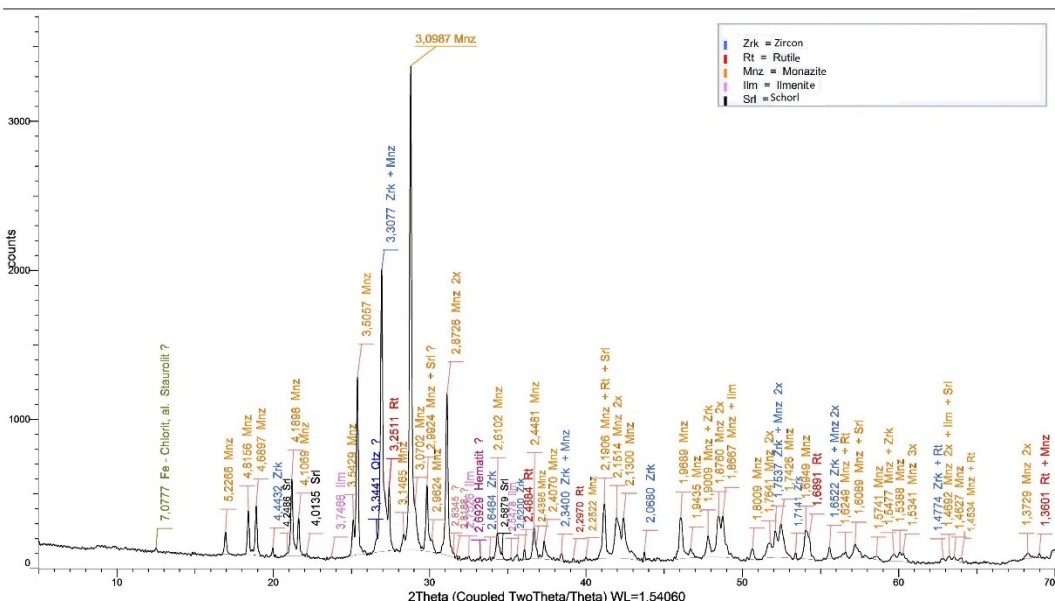

**Figure 5.** XRD diffractogram of monazite concentrate.

**Table 7.** Mineralogical composition of monazite concentrate from MS 0.5A.

| Mineral | wt % |
|---|---|
| Monazite | 86.6 |
| Rutile | 6.0 |
| Zircon | 5.5 |
| Hematite | 0.5 |
| Ilmenite | 0.5 |
| Other | 0.7 |

**Table 8.** Mineralogical composition of zircon concentrate.

| Mineral | wt % |
|---|---|
| Zircon | 99.0 |
| Rutile | 0.6 |
| Other | 0.4 |

The zircon concentrate contained 63.7% $ZrO_2$ and 1.34% $HfO_2$, which, at a stoichiometric content of 67.22% $ZrO_2$ in zircon [22], means that the zircon concentrate contained 96.1% zircon. According to the X-ray diffraction analysis, the concentrate contained up to 99% zircon.

The total amount of the concentrates of valuable heavy minerals obtained during the main experiment is shown on Table 9.

**Table 9.** Amount of VHM obtained during the main experiment.

| Mineral Concentrate | Amount |
|---|---|
| Zircon | 10 kg |
| Rutile | 10 kg |
| Ilmenite | 90 kg |
| Garnet | 750 kg |
| Monazite | 400 g |
| Gold | 3.5 g |

Photos of the obtained concentrates of individual minerals are shown in Figure 6.

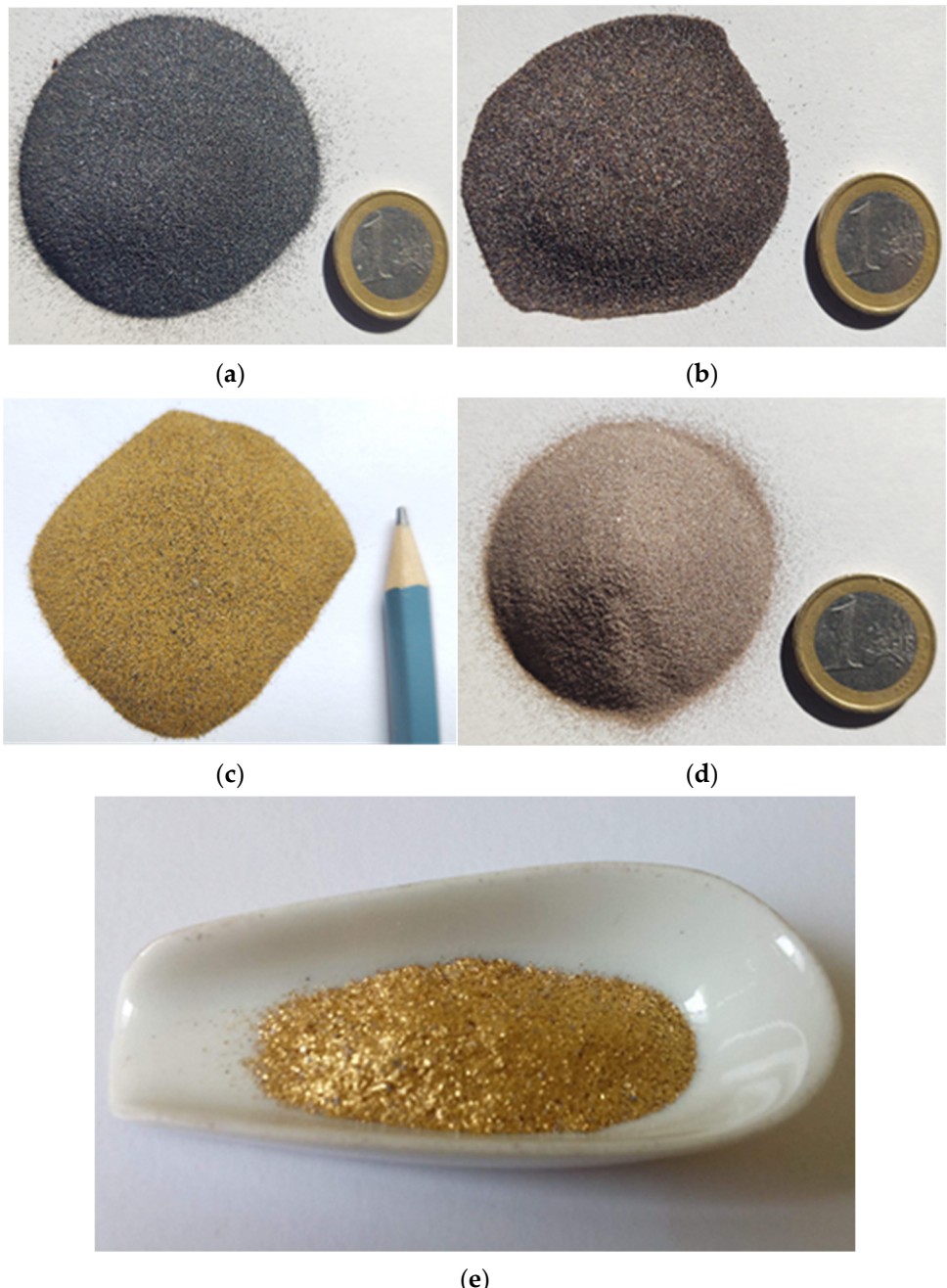

**Figure 6.** Photos of the prepared concentrates: (**a**) ilmenite, (**b**) rutile, (**c**) monazite, (**d**) zircon, (**e**) gold.

## 4. Conclusions

Experiments of gravity and electromagnetic separation were carried out on a sample of heavy mineral concentrate (obtained by sluicing) from gravel sands from the active river channel of the Danube River to obtain concentrates of industrial and ore minerals. Concentrates of magnetite, ilmenite, garnet, zircon, rutile, monazite and gold were prepared using the above separation methods.

It was shown experimentally and analytically that, through a combination of gravity and electromagnetic separation, concentrates of individual useful minerals (garnet, ilmenite, monazite, rutile, zircon, gold) can be successfully prepared from heavy mineral concentrates.

The extraction of VHM as a byproduct of extraction of the gravel sands (with annual rate of about $10^9$ tons) in the European Union can decrease the dependence on foreign imports of critical raw materials. The economy of this process depends on successful prepa-

ration of primary heavy mineral concentrates (Humphrey spirals, automatized sluices). The cost of obtaining heavy mineral concentrates in the way described above represents a fraction of the resources required to obtain the concentrates from primary industrial mineral and ore deposits, while the technological process of treatment can be simplified and streamlined. Since the gravitational separation on the concentration table took place on a working surface intended for medium- to coarse-grained materials, it is reasonable to assume that with the grain characteristics of heavy mineral concentrate, better quality parameters of most concentrates could be achieved using a working surface for fine-grained materials.

The volume of the alluvial coarse-grained sedimentary infill from the Slovakian part of the Danube basin was calculated as 169.6 km$^3$ [24]. With known contents of the heavy minerals 0.62 kg·t$^{-1}$ and gold 0.004 g·t$^{-1}$ gravel sand, the sedimentary infill contains approximately 210.5 million tons of heavy minerals and 1358 tons of gold [25]. With the annual volume of 600,000 cubic meters [26] influx of the coarse-grained sediments to the Slovakian part of the Danube basin, the Danube river gravel sand represents a practically endless source of heavy minerals.

**Author Contributions:** Conceptualization, M.M. (Michal Maťašovský); methodology, M.M. (Michal Maťašovský) and P.U.; software, P.U.; investigation, M.M. (Michal Maťašovský); resources, M.M. (Michal Maťašovský) and M.S.; data curation, M.M. (Michal Marcin); writing—original draft preparation, M.M. (Michal Marcin); writing—review and editing, M.M. (Michal Maťašovský) and M.S.; supervision, M.S. All authors have read and agreed to the published version of the manuscript.

**Funding:** This research received no external funding. This research was funded by the European Institute of Innovation and Technology (EIT), a body of the European Union, under the Horizon 2020, the EU Framework Programme for Research and Innovation (Project BioLeach: Innovative Bio-treatment of RM, grant number: 18259).

**Conflicts of Interest:** The authors declare no conflict of interest.

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
