# Peer review of "The Possibility of Separation of Heavy Minerals as Byproduct of the Danube River Gravel Sand Extraction"

_minerals, doi:10.3390/min12060659_

Round 1

Reviewer 1 Report

The Manuscript is a more extended and continuation of the pilot study (MaÅ¥ašovský, M., Šándorová, K., & Marcin, M. (2017). Danube river sandy gravels: New look on mineral resource potential. 320 International Multidisciplinary Scientific GeoConference: SGEM, 17(1.1), 819-823.). The authors recovered valuable minerals using physical separation methods and presented the flow chart for these processes. Although this study, in which gravity separation and magnetic separation methods are used, does not have a very high original value, it has the potential to contribute to the literature in the field of mineral processing by physical methods. The article can be accepted after corrections made. My opinion is to give a minor revision for this manuscript.

1-In the “Introduction” section, spelling adjustments are required. For example, the sentence “A result, heavy minerals such as monazite, rutile, ilmenite, are regarded important [2].” must be recast (should be “as a result”?).

2-Gold panning section should be added to the Figure 3.

3-In the last sentence of the “Conclusion” section, the word “represent” should be “represents”.

4- The manuscript should be thoroughly rechecked for typos.

5-I checked the manuscript using the ithenticate plagiarism detection program. The similarity rate was reasonable.

6- A few sentences about advanced separator devices (falcon concentrator, knelson concentrator, hydrocyclone, etc.) for the recovery of valuable contents from heavy mineral resources should be added to the Introduction section. The citations of the following references in these sentences will increase the value of the article.

-Altıner, M., Top, S., Kaymakoğlu, B. & Bayat, O. (2021). Dissolution of Uranium and Rare Earth Elements from a Low-Grade Phosphate Ore Using Different Acids . Geosound, 54 (1), 66-83. Retrieved from https://dergipark.org.tr/tr/pub/geosound/issue/66136/1035058.

-George Blankson Abaka-Wood, Keith Quast, Massimiliano Zanin, Jonas Addai-Mensah, William Skinner, A study of the feasibility of upgrading rare earth elements minerals from iron-oxide-silicate rich tailings using Knelson concentrator and Wilfley shaking table, Powder Technology, Volume 344, 2019, Pages 897-913, ISSN 0032-5910, https://doi.org/10.1016/j.powtec.2018.12.005.

-Hüseyin Vapur, Soner Top, Mahmut Altiner, Åžükrü Uçkun & Musa Sarikaya (2020) Comparison of iron ores upgraded with Falcon concentrator and magnetic separators assisted by coal reduction-conversion process, Particulate Science and Technology, 38:4, 409-418, https://doi.org/10.1080/02726351.2018.1548532.

-Haiyun Xie, Rui Sun, Xiangjun Ren, Zhicheng You, Yanhao Liu, Dongxia Feng, Luzheng Chen, Development of a novel fluidized hydrocyclone concentrator for mineral separation, Separation and Purification Technology, Volume 248, 2020, 116960, ISSN 1383-5866, https://doi.org/10.1016/j.seppur.2020.116960.

-Tonmoy Kundu, Surya Kanta Das, Sunil Kumar Tripathy, Shivakumar I. Angadi, Performance evaluation of the VSK separator for treating mineral fines, Minerals Engineering, Volume 167, 2021, 106883, ISSN 0892-6875, https://doi.org/10.1016/j.mineng.2021.106883.

Author Response

Thank you for your review. All of the points were corrected.

Reviewer 2 Report

The paper „The possibility of separation of heavy minerals as byproducts of the Danube River gravel sand extraction” is focused on non-chemical separation of minerals included in the Danube river deposit. In reference to current raw materials market it’s essential to find a new sources as well as new effective sourcing technologies. In my opinion, the paper meets this trend and is fit in the current needs in mineral processing. It should be interesting for the Minerals journal readers and I recommend it for publication after minor edition.

The paper composition and approach to the issue is proper in general. I have only few minor comments to the text:

  1. Lines 25-27; Opinion about ineffective recycling is too general. There are very effective recovery technologies for many elements. Development of recycling technologies is currently very dynamic. As it's opinion and there are no specific literature examples to the first two sentences, they should be edited or omitted.
  2. “extremely specific metals” it’s sufficient “specific metals”, without “extremely”
  3. There are some sequence disturbance in literature citation numbers – eg. line 33
  4. Line 85 - … ; It’s not clear why 0.315 grain size was a limit – please explain in text, why it was chosen as an optimal.
  5. Chemical analysis section needs some clarifications – I’m not sure of the translation “internal certified protocols” – do you mean “analytical procedures” or “accredited analytical procedures”? What is ME-MS88 protocol? – please discuss it with authors of analysis and give a citation or explain.
  6. You noted that a number of analytical methods were used, however the paper contain mostly XRD results. For description of approach and final results of your research it is sufficient, however it could be much more informative especially for further processing of  such materials if chemical analysis results for concentrated minerals will be presented. If it’s possible, and if you have such results, please add them as  supplementary materials.  
  7. Line 144: X-ray fluorescence spectrometry (XRF)
  8. General comment to XRD diagrams – there are plenty of signals and such presented data may be illegible in the final paper. This is of course important proof of the results, however the data should be easy to read. You may replace the diagrams with tables with corresponding data (it will be probably better) or edit the diagrams – split them and magnify. The diagrams in full scale could be also added as supplementary data.
  9. Figure 3 – Yes, it’s very important scheme presentation, but need some edition. There is probably lack of arrow to the concentrate. Some arrow descriptions need also to be centered.
  10. Tables 2-5 – is the MS 1A – magnetic separation 1 A? Please use the full name or clarify the abbreviations.
  11. Line 227 -231: REE are of course very interesting – as you analyze them there should be analytical data to determine specific elements – introduction of information about concentration of specific elements is essential for further processing.
  12. Figure 6: If it’s possible, please make photos with some scale ruler. It will be much more suitable for the publication, than coin or pencil.

Author Response

Thank you for your review. 

Reviewer 3 Report

The paper is written well. The idea is good to separate heavy minerals as by product with gold extraction from gravel sand. The separation procedure and methodology are clearly described. Mineral sand industry uses combination of gravity, magnetic and electrostatic separator but this paper describe  only gravity and electromagnetic separator to separate individual minerals which is unique.

In the methodology, it mentions about the chemical analyses such as ICP-MS, ICP-OES, AAS and TGA but in the result section there is no data of those analyses. The only single data REE is 51% in monazite concentrate but not in details. 

Further comments are attached "in comments on the article.docx"

Author Response

Thank you for your review. 
